

# Fusion rules in a Majorana single-charge transistor

**Rubén Seoane Souto⋆ and Martin Leijnse**

Division of Solid State Physics and NanoLund, Lund University, S-22100 Lund, Sweden
Center for Quantum Devices, Niels Bohr Institute, University of Copenhagen,
DK-2100 Copenhagen, Denmark

⋆ ruben.seoane_souto@ftf.lth.se

## Abstract

A demonstration of the theoretically predicted non-abelian properties of Majorana bound states (MBSs) would constitute a definite proof of a topological superconducting phase. Alongside the nontrivial braiding statistics, the fusion rules are fundamental properties of all non-abelian anyons. In this work, we propose and theoretically analyze a way to demonstrate MBS fusion rules in a Majorana single-charge transistor. Our proposal reduces both the number of operations and the device complexity compared to previous designs. Furthermore, we show that the fusion protocol can be adapted to pump a quantized amount of charge in each cycle, providing a straightforward method to detect fusion rules through a DC current measurement. We analyze the protocol numerically and analytically and show that the required operational timescales and expected measurement signals are within experimental capabilities in various superconductor-semiconductor hybrid platforms.

# 1   Introduction

There is currently a big interest in topological superconductors (TSs) with spinless $p$-wave pairing, fueled by the special properties that have been predicted for the Majorana bound states (MBSs) they host [1–7]. Two MBSs together encode a single fermionic state, fixed at zero energy if the MBSs remain well separated. The state is non-local in the sense that local measurements, *i.e.* coupling only to one MBS in the pair, cannot detect whether the fermionic state is empty or full. A system with $N$ well-separated MBSs has a $2^{N/2}$-fold degenerate ground state, where no local measurement can distinguish between the different ground states and no local perturbation can cause decoherence, unless it changes the particle number or involves excitations above the superconducting gap. This property provides topological protection against local noise sources. In addition, MBSs are non-abelian anyons, meaning that the exchange of two MBSs (so-called braiding) leads to nontrivial rotations within the degenerate ground state manifold [6,8–10]. This rotation is independent of the details of the exchange operation and thus allows error-free manipulation of the quantum state. The combination of protected storage and protected manipulation of quantum information forms the foundation for topological quantum computing schemes [11]. On a more fundamental level, the non-local and non-abelian nature of MBSs represents a fascinating new aspect of quantum physics that is unique to systems with a topological degeneracy.

Early theory works have suggested the possibility to engineer one-dimensional (1D) TSs by combining conventional superconductors with semiconductor nanowires with strong spin-orbit coupling [12, 13]. Experimental measurements have, for example, shown a zero-bias conductance peak consistent with the presence of MBSs at the ends of the 1D nanowire [14]. Later experiments have, for example, observed the expected scaling and quantization of the zero-bias peak [15], measured interferometry signatures [16], explored the hybridization with quantum dot orbitals [17, 18], measured the $4\pi$ Josephson effect [19], and studied the local and non-local transport in multiterminal wires [20, 21]. In Coulomb-blockaded nanowires, experiments have reported $1e$ periodicity as a function of the offset charge in the island [22–24], spin-split subgap states [25], and the exponential dependence of the subgap state energy on the island length [26, 27].

Despite the experimental effort, the demonstration of the non-abelian and/or non-local properties of MBSs remains an open challenge in the field and, in fact, one of the most heavily pursued goals in condensed matter physics today [6]. Without such a demonstration, it seems impossible to distinguish with certainty topological MBSs from different types of non-topological states, that might also explain many of the experimental observations so far, including the conductance measurements, see for example Refs. [7,28–35]. Some of these trivial states may also reproduce other properties associated to topological Majoranas, including the $4\pi$ Josephson effect [31, 36].

The exchange of MBSs in real space [10, 37, 38] seems challenging as it would require moving the boundaries between topological and trivial regions, which has not yet been experimentally demonstrated up to now. For this reason, proposals have appeared based on different ways to effectively exchange MBSs in parameter space. Some examples include tuning the coupling between MBSs in time [39–43], repeated measurements of different MBS

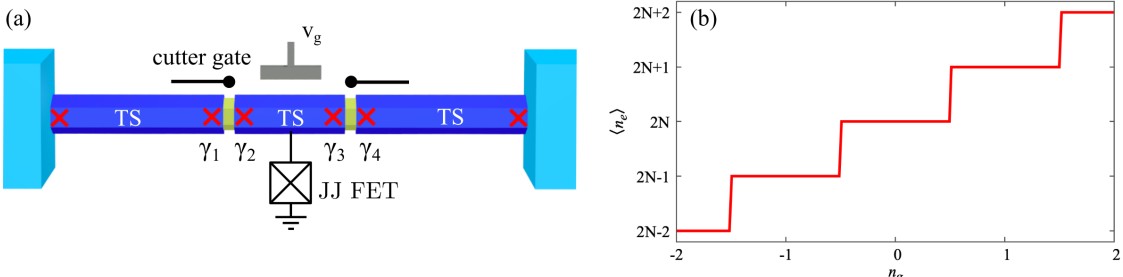

Figure 1: (a) Device sketch, where light (dark) blue are trivial (topological) superconductors and the red crosses represent MBSs. The black lines represent cutter gates, tuning the coupling of the central island to the other two TSs and $V_g$, proportional to $n_g$, tunes the island occupation. The island is connected to a JJ FET used to quench charging energy in the device. (b) Charge in the ground state of the island, $\langle n_e \rangle$, as a function of the charge offset $n_g$ in units of the electron charge. In the isolated case, the island ground state charge increases in a discrete way when $n_g$ is varied.

pairs [44–47], or shuttling single electrons between quantum dots and TSs [48–50]. Alternatively, non-abelian properties can be shown in 2D TSs by injecting vortices in the edge channel [51–53].

An alternative to braiding would be to demonstrate the so-called fusion rules, which are a related and equally fundamental property of non-abelian anyons. Both fusion rules and non-abelian braiding arise from the topological ground-state degeneracy. Generally, fusion means that bringing together two (or more) non-abelian anyons can result in multiple types of other anyons. MBSs are an example of Ising anyons, denoted by $\sigma$. The MBS fusion rule can then be written

$$\sigma \times \sigma = I + \psi, \tag{1}$$

which means that two Ising anyons can fuse into either the trivial particle/vacuum (indicated by $I$) or into a regular fermion (indicated by $\psi$). A less formal but equivalent way of viewing this is that the fusion (coupling and measurement) of two MBSs has two possible outcomes corresponding to the occupation of the fermion spanned by the measured MBS pair: empty, related to $I$, and full, related to $\psi$. The probability for each outcome depends on the joint state of the two MBSs. To experimentally demonstrate the fusion rules, one needs to initialize the system such that certain MBS pairs have a well-defined state and then measure the MBSs paired differently. Similar to braiding, the result is topologically protected and insensitive to the details of the measurement. Different methods have been proposed to demonstrate the MBS fusion rules [41, 54–61].

In our work, we propose a Majorana single-charge transistor as a platform to demonstrate MBS fusion rules (Fig. 1 (a)). The device is composed by a TS island, whose ground-state charge can be tuned using electrostatic gates (Fig. 1 (b)). The island couples to a grounded bulk superconductor through a tunable Josephson junction (JJ) field electron transistor (FET). When the JJ FET is open, the island can exchange Cooper pairs with the bulk superconductor, quenching charging effects in the island (see Refs. [62–64] for experimental realizations of this technology). When the JJ FET is closed, no Cooper pairs can be transferred, making the island ground state non-degenerate. Finally, the island couples to two grounded TSs.

Our proposal shares some elements with the one in Refs. [41,57] but offers several advantages: (1) it requires only one topological island and one JJ FET instead of two; (2) in addition to initialization and readout, it requires only two steps to perform fusion instead of three; (3)

The outcome of the fusion protocol can be read out by measuring the charge pumped between the JJ FET and the TS leads, leading to a DC current when the protocol is done periodically; (4) the geometry in Fig. 1 (a) has already been investigated in semiconductor-superconductor platforms [65–68].

Using a low-energy model, we set bounds on the fusion timescales. For typical parameters in experimentally available superconductor-semiconductor hybrid structures, we estimate that pulses have to be slower than 100 ps. The upper limit is determined by how well we can switch off the coupling to the TSs and by the maximal achievable coupling to the grounded SC via the JJ FET, as well as by overlaps between the island MBSs and quasiparticle poisoning. We expect these timescales to be within the capabilities of standard arbitrary waveform generators and to leave plenty of room for varying the pulses to demonstrate topological protection of the fusion outcome. During the fusion protocol, the ground state of the island is doubly degenerate, leading to a parity superposition that can be measured as an excess charge in the island 50% of the time using standard charge-detection techniques [69]. We also design a reference experiment where the ground state of the island remains non-degenerate throughout the protocol, leading to a state with a well-defined charge in the island. Finally, we propose a modified cyclic protocol where the excess electron, which arises with 50% probability for each fusion cycle, leads to a pumped charge between the TS leads and the JJ FET. The parameters can be adjusted to make the pumped charge quantized. This cyclic protocol allows the detection of MBS fusion rules based on fast control but slow (DC) measurements. Throughout the paper we use units where $\hbar = e = 1$.

## 2 Fusion protocol

In this section, we give an intuitive explanation of the fusion protocol. The detailed model and accompanying discussion about the energy spectrum and related timescales are presented in Section 3. The proposed fusion protocol is schematically shown in Fig. 2. To ensure a unique ground state of the isolated island, it has to be tuned to a Coulomb valley where the ground state charge is well-defined (Fig. 1(b)). In the following, we will, without loss of generality, assume that $n_g$ is tuned such that the number of electrons in the ground state of the isolated island is even.

The model Hamiltonian will be discussed in Section 3; here it is enough to introduce the Fock space spanned by pairing the four inner MBSs, $\gamma_1 - \gamma_4$ in Fig. 1(a), to form two fermions. We denote by $f_{mn}^\dagger = \gamma_m - i\gamma_n$ the creation operator of a fermion with occupation operator $n_{mn} = f_{mn}^\dagger f_{mn}$, such that the state spanned by the four MBSs can be written as $|n_{mn}n_{kl}\rangle$. Different ways to pair MBSs (*i.e.* different ways to choose $mn$ and $kl$) correspond to different choices of basis. In the fusion protocol, we will prepare the system in one basis and then measure it in another.

The device allows coupling MBSs in two different ways. If, say, the leftmost cutter gate is open, MBSs $\gamma_1$ and $\gamma_2$ overlap which results in a finite energy splitting between the state with $n_{12} = 0$ and the state with $n_{12} = 1$. If both cutter gates and the JJ FET are closed, the charging energy associated with the island effectively couples $\gamma_2$ and $\gamma_3$, resulting in a finite energy splitting between the states with $n_{23} = 0$ and $n_{23} = 1$ states (see Section 3 for details). Closed cutter gates and open JJ FET turns off all interactions between MBSs and leads to a four-fold degenerate ground state, $|n_{12}n_{34}\rangle$ with $n_{12}, n_{34} = 0, 1$.

The protocol begins in the top panel of Fig. 2, where the island is coupled to both TSs and the JJ FET, quenching the effect of charging energy on the island. The coupling between MBSs in the island and the leads provides an energy splitting between states with different $n_{12}$ and $n_{34}$. In this situation, the global ground state is non-degenerate. By waiting long enough,

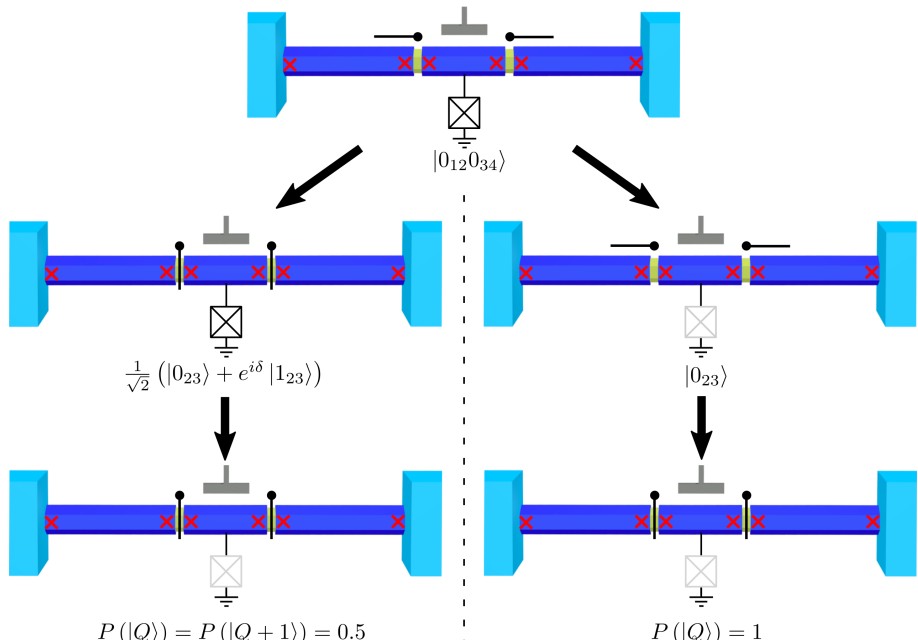

Figure 2: Fusion (left) and reference (right) protocols. Initially, the island is connected to the leads and charging effects are quenched by opening the JJ FET. In the fusion protocol, the cutter gates are first closed (indicated by the black vertical lines cutting the TS wires), and then the JJ FET is closed (indicated by a faded symbol). In the reference protocol, the JJ FET is closed before the cutter gates. At the end, the island is isolated. In the reference protocol the isolated island always ends up in the ground state (charge $Q$). In the fusion protocol the isolated island ends up in either the ground state (charge $Q$) or first excited state (charge $Q + 1$) with equal probabilities.

the system is initialized in the ground state which, without loss of generality, we assume to be $|i\rangle = \left|0_{12}0_{34}\right\rangle$.

The fusion protocol is sketched in the left panels of Fig. 2. We now switch off the coupling to the TS leads, leading to a four-fold degenerate ground state. If the operation is done adiabatically and without generating single quasiparticle excitations, the values of $n_{12}$ and $n_{34}$ must remain the same. We finally close the JJ FET, isolating the island from the reservoirs. Again, this action cannot change the values of $n_{12}$ and $n_{34}$. The result of the fusion protocol is more clear by changing the MBS basis to 1&4 and 2&3

$$\left|0_{12}0_{34}\right\rangle \longrightarrow \frac{1}{\sqrt{2}} \left( \left|0_{23}0_{14}\right\rangle + e^{i\delta} \left|1_{23}1_{14}\right\rangle \right), \tag{2}$$

where $\delta$ is a phase that includes dynamical contributions due to the energy splitting between the $n_{23} = 0$ and $n_{23} = 1$ states induced by switching on the Coulomb blockade on the island. This phase plays no role in the protocol, in fact we expect the superposition in Eq. (2) to quickly decohere into a classical equal-probability mixture of $\left|0_{23}0_{14}\right\rangle$ and $\left|1_{23}1_{14}\right\rangle$. Importantly, the charge on the island depends on the value of $n_{23}$. Let's assume that the charge is $Q$ for $n_{23} = 0$ and $Q + 1$ for $n_{23} = 1$. At this point, the charge in the island can be measured using charge sensing methods [69]. The 50% chance for measuring $Q$ and $Q + 1$ respectively is a manifestation of the MBS fusion rule. The fact that the outcome is robust to changes of the pulse strengths and durations, drifts on the ground state charge, and other details of the protocol (within some limits, see Section 3) shows the topological nature of the fusion process.

As a further verification that the outcome of the protocol outlined above is due to the non-trivial fusion rules, we propose a reference protocol that gives a trivial result. In the reference protocol, the starting and end points, as well as the intermediate steps, are the same as in the fusion protocol. However, the intermediate steps are performed in the reversed order. The reference protocol is described in the right panels of Fig. 2. We begin again with the island coupled to the leads and the JJ FET open (top panel in Fig. 2), giving the same initial state $|i\rangle = |0_{12}0_{34}\rangle$ as in the fusion protocol because of the couplings between MBSs 1&2 and 3&4. Closing the JJ FET adiabatically with the couplings to the TSs still on, introduces a charging energy which couples also MBSs 2&3. As a result, the state evolves into some superposition of $|0_{12}0_{34}\rangle$ and $|1_{12}1_{34}\rangle$. The weights in the superposition depend on the relative coupling strengths and are irrelevant; what is important is that this state is non-degenerate. Finally, the coupling to the TS leads is switched off adiabatically which makes the state evolve into $|0_{23}0_{14}\rangle$, corresponding to the islands ground state charge, $Q$. The key here is that, in contrast to the fusion protocol, the ground state stays non-degenerate throughout the reference protocol, leading to the same outcome every time. For that, it is important to stay in the same Coulomb valley when closing the coupling to the TS leads.

Alternatively, MBS fusion rules can be demonstrated by running the two protocols in reverse: starting from a closed island (last step in Fig. 2), and going towards a fully open configuration. In this case, the system is initialized in a state with a well-defined parity of MBSs 2&3, evolving into a situation where MBSs 1&2 and 3&4 couple. In this case, the parity of MBSs 1&2 and 3&4 determine the supercurrent at the left and the right interface [1]. This would allow measuring the outcome of the protocols through supercurrent measurements, which have to be performed faster than the system relaxation timescale.

In the following sections, we introduce a minimal model to describe our system and discuss the limiting timescales for the operations required for the fusion protocol. The limiting timescales for the reference protocol are obtained in the Appendix A.

## 3   Model Hamiltonian

To estimate bounds on the fusion timescales, we use an effective low-energy model given by

$$H = H_l + H_I + H_T + H_J \, . \tag{3}$$

We assume vanishing overlaps between MBSs in the leads, $H_l = 0$. We take the limit where the superconducting gap is larger than any other energy scale in the model of Eq. (3). The Bogoliubov operators are given by $\gamma_1 = f_{14} + f_{14}^\dagger$ and $\gamma_4 = i(f_{14} - f_{14}^\dagger)$, where $f_{14}$ is the annihilation operator of the non-local fermion formed by the left and right MBSs in the leads, Fig. 1. The island Hamiltonian is given by

$$H_I = -i\varepsilon_I \gamma_2 \gamma_3 + E_C (n - n_g)^2 \, , \tag{4}$$

where $\gamma_3 = f_{23} + f_{23}^\dagger$ and $\gamma_2 = i(f_{23} - f_{23}^\dagger)$ are the Bogoliubov operators for the MBSs in the island. Here, $E_C$ is the charging energy, $n$ is the number of electrons on the island (including Cooper pairs) relative to some arbitrary reference charge, and $n_g$ describes the effect of an electrostatic gate, tuning the island charge. The energy of the non-local fermionic state in the island, $\varepsilon_I$, is taken to be zero in the calculations. The tunneling Hamiltonian is given by

$$H_T = E_M \cos\left(\frac{\phi_L - \phi_I}{2}\right)\gamma_1 \gamma_2 + E_M \cos\left(\frac{\phi_R - \phi_I}{2}\right)\gamma_4 \gamma_3 \, , \tag{5}$$

where $\phi_\nu$ are the superconducting phase of the lead $\nu = L, R$. The operator $\phi_I$ is the conjugate operator to the electron number operator $n$ in the island, implying that $e^{-i\phi_I}$ destroys a Cooper

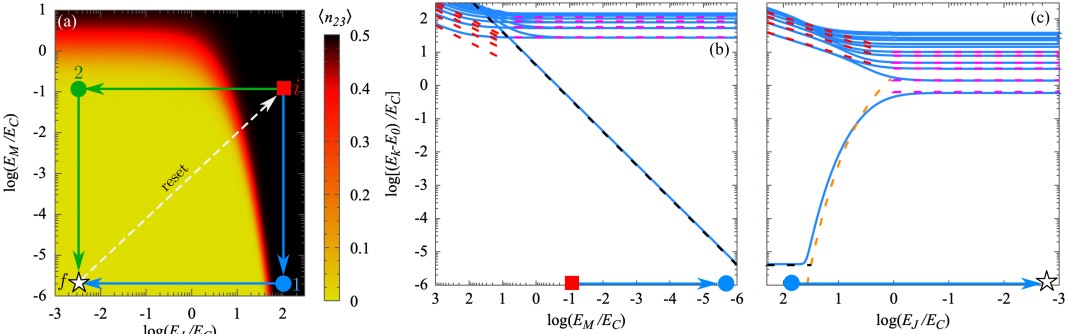

Figure 3: (a) Island MBSs parity in the ground state as a function of $E_J$ and $E_M$. The red square represents the initial configuration ($i$) where the island is coupled to the TS leads and the JJ FET is open. The blue (green) arrows correspond to the fusion (reference) protocol, where the dot is an intermediate step where the island is decoupled from the bulk superconductor (TS leads). In the final configuration, white star ($f$), the island is isolated. Panels (b) and (c) represent the excitation energies from the ground state in the total even parity subspace when decoupling the TS leads and the JJ FET, following the fusion path. The same energies are found for the total odd parity subspace. Solid lines correspond to numerical results while the dashed lines are approximations in the different regimes (details are given in the text). We use $E_J^{\max} = 100 E_C$, $E_J^{\min} = E_C/100$, $E_M^{\max} = 0.4 E_C$, $E_M^{\min} = 10^{-6} E_C$, and $n_g = 0.2$.

pair. For simplicity, we consider that the island couples symmetrically to the leads, although the results do not change qualitatively for asymmetric couplings. The tunneling between distant MBSs (1&3, 2&4, and 1&4) is strongly suppressed and is neglected in the calculations. In principle, Eq. (5) can contain also a standard Josephson term describing tunneling of Cooper pairs, [70, 71]. For weak tunnel couplings, this term is much smaller than the ones in Eq. (5), because it scales with the square of the tunnel coupling. In this work, we have neglected this term to obtain conservative analytic estimates for the lower timescale in the designed protocols (see below). If it is included, it adds to $H_J$, increasing the energy splitting between the Hamiltonian eigenstates, reducing the lower timescale limit. Finally, $H_J = E_J \cos \phi_I$ describes the exchange of Cooper pairs between the superconducting ground and the island through the JJ FET, controlling the effect of the charging energy.

The calculations we performed are based on diagonalizing the Hamiltonian in Eq. (3), written in the charge basis. It provides the eigenenergies, the island charge, and the MBS occupation. We use a total of 120 island charge states, which is enough for convergence, and offset $n_g$ such that $n_g = 0$ is at the middle of the considered charge interval.

In Fig. 3 (a) we show the ground state parity of the island MBSs, $\langle n_{23} \rangle = \langle f_{23}^\dagger f_{23} \rangle$. The red square represents the initial point for the protocol, see top panel of Fig. 2. The blue and green arrows denote the paths for the fusion and reference protocols. In order to initialize the system in a state with well-defined parity of 1&2 and 3&4 MBSs, we require $E_J^{\max} \gg E_C$, so charging energy is quenched when the JJ FET is open, and $E_J^{\max} \gg k_B T$. We also consider $E_M^{\max} \ll E_C$, but the protocol does not rely on this assumption. The blue/green dots in Fig. 3(a) correspond to the intermediate steps for the fusion/reference protocols in Fig. 2, while the white star is the final point where measurements are performed.

# 4 Fusion timescales

Our estimate for the fusion timescales is based on the adiabatic condition, required for the system to remain in its ground state during the full protocol. This condition should be fulfilled throughout the fusion and reference protocols, except close to point 1 in Fig. 3 (a), where the ground state is close to doubly degenerate and the operation should be non-adiabatic with respect to the small remnant splitting. The adiabatic condition is given by [72]

$$\max_k [f_k(t)] \ll 1 , \tag{6}$$

where

$$f_k(t) = \frac{|\langle \psi_k(t)|\partial H(t)/\partial t|\psi_0(t)\rangle|}{[E_k(t) - E_0(t)]^2} . \tag{7}$$

Here, $|\psi_k(t)\rangle$ are the even total parity eigenstates with $k = 0$ being the ground state, and $E_k$ the corresponding eigenenergies. Eq. (7) can be turned into a condition on the operation timescale $\Delta t$ as [57]

$$\Delta t \gg \max_k \int d\lambda \frac{|\langle \psi_k(\lambda)|\partial H(\lambda)/\partial \lambda|\psi_0(\lambda)\rangle|}{[E_k(\lambda) - E_0(\lambda)]^2} , \tag{8}$$

where $\lambda(t)$ describes the parameter paths shown in Fig. 3 (a). Below, we give conservative estimates for the limiting timescales by choosing the maximal value for the numerator in Eq. (8), $|\langle \psi_k(\lambda)|\partial H(\lambda)/\partial \lambda|\psi_0(\lambda)\rangle| = 1$ for the undesired transitions to the lowest-energy excited state. Therefore, our estimates for the limiting timescales are based on the excitation energies of the low-energy Hamiltonian in Eq. (3). This simplification allows us to obtain analytic expressions for $E_k$ and bounds for the operation timescales in the different regimes.

## 4.1 Step 1: Inducing a ground state degeneracy

At the starting point, $i$ in Fig. 3 (a), the protocol requires MBSs 1&2 and 3&4 to have a well-defined parity. This condition is achieved when the ground state at $i$ is separated from the excited states by an energy much larger than the thermal energy, given by $4E_M^{\max} \gg k_B T$ for $E_J^{\max} \gg E_C$. In this limit, the system will relax to the global ground state. The initial parities of the 1&2 and 3&4 pairs depend on the superconducting phase difference $\phi_L - \phi_R$.

In the first step of the protocol ($i \rightarrow 1$ in Fig. 3 (a)), the island decouples from the leads, reducing $E_M$. The reduction in couplings between MBSs 1&2 and 3&4 can be seen as an excitation energy approaching zero in Fig. 3 (b), which is described by

$$E_1 - E_0 = 4E_M , \tag{9}$$

shown by the black dashed line. However, even in the limit $E_M \rightarrow 0$ (not shown in Fig. 3), $E_1 - E_0$ remains finite. One contribution is given by the overlap between MBSs in the island, $\varepsilon_I$. But even for $\varepsilon_I \rightarrow 0$ the fact that a finite $E_J$ does not completely quench charging effects induces a splitting between the ground and excited states which is well approximated by [40, 57]

$$E_1 - E_0 = \frac{32}{(2\pi^2)^{1/4}} \left(E_J^3 E_C\right)^{1/4} e^{-\sqrt{8E_J/E_C}} , \tag{10}$$

with the minimum value given by $E_J = E_J^{\max}$. The protocol should be non-adiabatic with respect to the lowest-energy excited state to ensure that the parity of the island is not well-defined at 1. Otherwise, the charge in the island evolves to its global ground state at the end of the fusion protocol, similar to the reference protocol, and the proposed fusion experiment

fails. Therefore, the maximum value of $1/(E_1 - E_0)$ determines an upper bound for the fusion timescale.

The lower bound for the operation timescale is given by the inverse excitation energies to the other states, according to Eq. (8). The limit $E_M > E_J^{\max} \gg E_C$ is dominated by the Josephson plasma oscillations and the excitation energies are given by

$$E_k - E_0 = k\sqrt{8E_M E_C}, \tag{11}$$

shown by the dashed red lines in Fig. 3 (b). In the opposite limit, $E_M < E_J^{\max}$, the excitation energies are described by

$$E_k - E_0 = k\sqrt{8E_J^{\max} E_C}, \tag{12}$$

shown by the dashed magenta lines in Fig. 3.

The analytic solutions to the excitation energies in these limits help us estimate a lower bound to manipulation timescales for the $i \to 1$ step. Using Eq. (8) we find

$$\Delta t_{i\to 1} \gg \frac{\log\left[E_M^{\max}/\min\left(E_J^{\max}, E_M^{\max}\right)\right]}{8E_C} + \frac{\min\left(E_J^{\max}, E_M^{\max}\right)}{8E_J^{\max} E_C}. \tag{13}$$

Also, the upper timescale limit for the operation is determined by the energy splitting between the ground and the excited state, given by

$$\Delta t_{i\to 1} \ll \int \frac{dE_M}{(E_1 - E_0)^2}, \tag{14}$$

where we have used $|\langle \psi_0(t)|\partial H/\partial E_M|\psi_1(t)\rangle| = 1$, as the ground and the first excited state differ in one charge in the island. Using Eq. (9) we find

$$\Delta t_{i\to 1} \ll \frac{1}{16E_M^{\min}}, \tag{15}$$

in the regime $E_M^{\min} \ll E_M^{\max}$.

## 4.2 Step 2: Lifting ground state degeneracy

In the second step of the protocol ($1 \to f$ in Fig. 3 (a)), the coupling between the island and the grounded superconductor mediated by the JJ FET is quenched, reducing $E_J$. It makes charging effects important in the island, providing an effective coupling between $\gamma_2$ and $\gamma_3$ and lifting the ground state degeneracy. It can be seen by the energy increase of the lowest-energy excited state in Fig. 3 (c). The energy increase is well-approximated by Eq. (10), represented by the orange line in Fig. 3 (c). Ideally, the fusion protocol results in an equal superposition between the ground and the excited states. Therefore, the system evolution in step 2 can be non-adiabatic with respect to this lowest excitation energy.

However, the evolution has to be adiabatic with respect to the higher excitation energies. In the limit $E_J \gg E_C$, the excitation energies are described by the Josephson plasma oscillation

$$E_k - E_0 = k\sqrt{8E_J E_C}, \tag{16}$$

red lines in Fig. 3 (c). In the decoupled limit ($f$ in Fig. 3 (a)), the excitation energies correspond to those of the island Hamiltonian $H_I$,

$$E_k = E_C\left(k - \{n_g\}\right)^2, \tag{17}$$

denoted by magenta lines in panel (c). Here, $\{n_g\}$ denotes the deviation from the center of the closest Coulomb valley, $-1/2 < \{n_g\} < 1/2$.

The dominant non-adiabatic errors are given by the transitions from the first to the higher excited states, changing the electron number in the island. A non-adiabatic closing of the JJ FET leads to different possible results of the fusion protocol at $f$, making it indistinguishable from the outcome of the reference protocol. For this reason, the closing down of the JJ FET has to be done adiabatically. The limiting timescale for the second step in the fusion protocol is given by

$$\Delta t_{1\to i} \gg \frac{1}{8E_C} \log\left(E_J^{\max}/E_C\right) + \frac{1}{E_C\left[(2-\{n_g\})^2 - (1-\{n_g\})^2\right]^2} \ . \tag{18}$$

The first term in the equation relates to the regime $E_J \gg E_C$, where the excitation energy between the first and the second excited states is mainly determined by the Josephson plasma energy, (16). The second term in Eq. (18) relates to the regime $E_J \lesssim E_C$. This contribution to the timescale of step 2 is minimal for integer $n_g$, where it is given by $1/E_C$.

## 4.3 Estimating protocol timescales

We can estimate a lower bound for the fusion timescale using Eqs. (13,18) as $\Delta t_{\text{fusion}} = \Delta t_{i\to 1} + \Delta t_{1\to f}$. For illustration purposes, we choose a charging energy of $E_C = 50$ μeV and take the same parameters used in Fig. 3. In this regime of parameters, the first term in Eq. (13) vanishes, while the second one provides a very low bound to the timescale $\Delta t_{i\to 1} \gg 10^{-15}$ s. The limiting timescale in the fusion protocol is therefore determined by the second step in the fusion protocol, Eq. (18). Using the parameters mentioned above, we find that $\Delta t_{1\to f} \gg 10^{-11}$ s, which is the estimated lower bound for the full fusion protocol. Drifts in $n_g$ when tuning $E_J$ does not affect the experiment outcome and the estimated timescales. However, it is important to stay in the same Coulomb valley while switching off $E_M$ to ensure the convergence of the reference protocol to a state with a well-defined island charge.

The limiting timescale for the reference protocol is analyzed in appendix A. For the parameters mentioned above, we find a lower bound for the reference protocol of $\Delta t_{\text{ref}} \gg 10^{-10}$ s. This limit is controlled by the energy splitting between the ground and the excited states when closing the JJ FET, see Appendix A. This splitting is controlled by $E_M^{\max}$. A higher Majorana coupling compared to the one used here reduces the lower bound for the reference protocol, at a cost of increasing $\Delta t_{fusion}$. The optimal operation timescale is found for $E_J^{\max} \approx E_M^{\max} >> E_C$, where $\Delta t_{fusion} \approx \Delta t_{ref}$. However, we note that our model is only valid for $E_M^{\max} < \Delta$.

The energy splitting between the ground and the excited states at point 1 in Fig. 3 (a) determines an upper bound for the fusion timescale. For the parameters chosen, $\Delta t_{i\to 1} \ll 10^{-5}$ s. The overlap between the MBSs in the island increases the energy splitting at 1. As $E_M$ has to be switched-off non-adiabatically with respect to the first excited state, this MBS overlap will contribute to decreasing the upper timescale limit for the $E_M$ switch off. In addition, the coupling between the MBSs in the same lead can change the total parity of the inner MBSs. For this reason, the experiment should be faster than the characteristic tunneling timescale between MBSs in the same lead, avoiding uncontrolled parity changes.

As discussed in section 4.1, there might be additional contributions to upper limit of the protocols timescales. The excitation of single electrons to the island MBSs, so-called quasiparticle poisoning, provides the lifetime of the island parity state, giving an additional limiting timescale. The poisoning timescale depends exponentially on the superconducting gap. For this reason, the fusion experiment needs to have TSs with a relatively large superconducting gap. In experiments, there is another limiting timescale related to the excitation of quasiparticles above the superconducting gap with an energy $2\Delta$. In the fusion protocol, the excitation of quasiparticles in the island during the JJ FET closing does not change the outcome of the

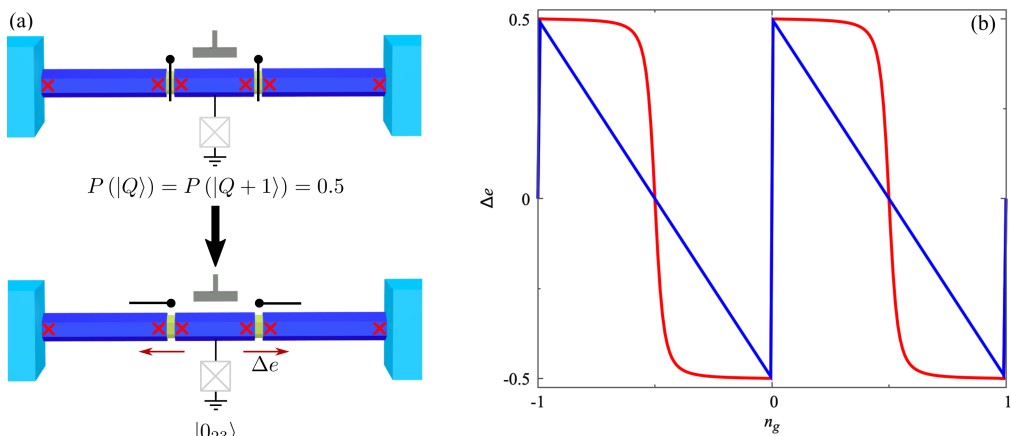

Figure 4: (a) Pump protocol where the coupling between the TSs is switched on after the fusion protocol (Fig. 2). At the end of the cycle, some charge can flow between the island and the TS leads (red arrows). (b) Number of electrons pumped out through the JJ FET per fusion cycle for $E_M^{\max} = 0.1 E_C$ (red) and $E_M^{\max} = 10 E_C$ (blue).

protocol. However, the excitation of quasiparticles in the system can be detrimental for the experiment when quenching $E_M$, as well as when quenching $E_J$ in the reference protocol. The timescale associated with these processes is given by $\Delta t_{\text{fusion}}, \Delta t_{\text{ref}} \gg E_J^{\max}/(2\Delta)^2$, illustrating the importance of developing clean materials with large and hard superconducting gaps [73–77]. Other trivial subgap states with energy $\leq \Delta$ can further increase the lower bound for the protocols. For this reason, it is important to tune the device in a regime where trivial states do not approach low energies during the protocol.

# 5 Detecting fusion through charge pumping

The charge on the island provides a way of determining the outcome of the fusion protocol, which can be measured using charge sensing on the island [69]. We present here an alternative way to probe the Majorana fusion rules which only relies on measuring a DC current.

In this scheme, to measure the outcome of the fusion protocol, we propose opening first the coupling to the TS leads after the fusion protocol (Fig. 4 (a)). This corresponds to following the blue line in Fig. 3 (a) between $i \to f$ and returning to the initial point through the green line $f \to i$. In this way, the excess charge, present on the island with 50% probability after fusion, is transferred to the TS leads in a timescale set by the relaxation of the island TS. Periodic loops in the $E_J$, $E_M$ plane will lead to a net current pumped between the grounded SC and the TSs, proportional to the frequency. It is important to note that no current is obtained if the system does not approach point 1 in Fig. 3 (a) in its adiabatic evolution, as the ground state would then remain non-degenerate.

The pumped charge between the JJ FET and the TS leads per cycle $\Delta e$ is given by difference in the island charge at $f$ and 2 in Fig. 3 (a), details are given in the Appendix B. In Fig. 4 (b) we represent $\Delta e$ as a function of the offset charge, $n_g$. In the limit where $E_M^{\max} \ll E_C$, $\Delta e$ shows quantized plateaus of $e/2$ resulting from the equal superposition between different parity states of the island MBSs at $f$. Transitions to the higher excited states and quasiparticle poisoning would lead to a reduction of the quantized value, being a measurement of undesired effects in the device.

The width of the quantized plateaus is reduced when increasing $E_M^{\max}$. In the limit

$E_M^{\max} \gg E_C$, the coupling to the TS leads quench charging effects in the island. In this limit, the charge at 2 and $i$ is equal (see Appendix B), making the plateaus disappear.

Using the realistic numbers obtained in Sec. 4.3, where fusion and reference protocols can be done in $\sim 10^{-10}$ s for $E_J^{\max} \gg E_C \gg E_M^{\max}$, we find that the maximum current obtained from the protocol is of the order of $\sim 1$ nA. Our estimate disregards, however, the system relaxation time at 2, where the excess charge tunnels out between the island and the TS leads. This process requires energy dissipation, determined by relevant environment degrees of freedom, such as phonons or the electromagnetic environment. Therefore, the estimated current corresponds to the case where the relaxation time $\tau_{relax} \ll \Delta t_{2 \to f}$, see Appendix A for the reference protocol timescales. In the opposite limit, the current will be reduced and determined by $1/2\tau_{relax}$.

## 6 Conclusions

In this work, we have shown the possibility of demonstrating Majorana fusion rules in a Majorana single-charge transistor geometry. Compared with previously suggested fusion protocols, our proposal requires a simplified device design and fewer steps. Readout in the experiment can be done via standard charge detection methods in the island.

Using a low-energy model, we have obtained bounds for the operation timescale to show Majorana fusion rules. The lowest timescale is determined by the transitions to the excited states. Using realistic parameters, we estimate the timescale for the fusion and reference protocol to be $\Delta t \gg 10^{-10}$ s.

The fusion protocol exploits the ground state degeneracy in the configuration where the island couples only to a grounded superconductor. For the paramaters used in our calculations, we find the bound $\Delta t_{i \to 1} \ll 10^{-5}$ s, limited by a remnant coupling between island and lead MBSs. There are other factors which might lower this timescale bound, such as a coupling between the island MBSs. The upper timescale is also limited by quasiparticle tunneling (poisoning), changing system parity in an uncontrolled way. As an alternative to single-shot measurements of the island charge, we show that the protocol can be carried out in a way such that the fusion process pumps a quantized charge between the TSs and the grounded SC. This allows a fusion-rule detection via measurement of a DC current.

We note that some trivial states can exhibit a full range of topological properties, including $4\pi$ Josephson effect and non-abelian exchange properties [31,36]. Our proposal cannot distinguish between topological MBSs and these trivial non-abelian states. Similar limitations apply to other fusion and braiding proposals.

An experimental demonstration of fusion rules, intimately related to non-abelian properties, will constitute a leap forward in the field, demonstrating a new kind of quasiparticle. It will open the door to applications, including protected quantum processing.

## Acknowledgements

We acknowledge Jens Schulenborg and Serwan Asaad for useful discussions and feedback on the manuscript. This project has received funding from the European Research Council (ERC) under the European Union's Horizon 2020 research and innovation programme under Grant Agreement No. 856526, QuantERA project 2D hybrid materials as a platform for topological quantum computing", and NanoLund.

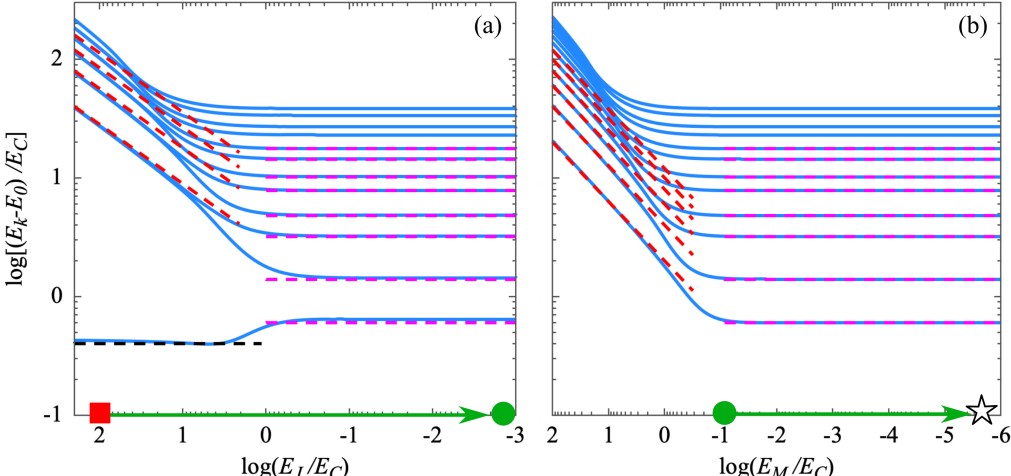

Figure 5: Excitation energies during the reference protocol. Parameters are the same as in Fig. 3.

# A    Reference protocol

In this section we obtain the limiting timescales for the reference protocol, shown in the right panels of Fig. 2 and by the green arrows in Fig. 3 (a). In this protocol, the JJ FET is first closed, as shown in Fig. 5 (a). The energy of the first excited state is described by

$$E_1 - E_0 = 4E_M \, , \tag{19}$$

for $E_J \gg E_C$, black line in Fig. 5 (a). In this limit, the higher excited states are described by the Josephson plasma energy

$$E_k - E_0 = k\sqrt{8E_J E_C} \, , \tag{20}$$

shown as red lines. In the limit where $E_J, E_M \ll E_C$ the excitation energies are well-described by the charging energy

$$E_k = E_C \left(k - \{n_g\}\right)^2 \, , \tag{21}$$

magenta lines in Fig. 5 (a). In the opposite limit where $E_M \gg E_C \gg E_J$, Josephson energy dominates and we recover Eq. (11) for the excitation energies.

In the final step of the reference protocol, the coupling between the island and the TS leads is switched off, leaving the island isolated from the environment. The evolution of the excitation energies is shown in Fig. 5 (b). The limit $E_M \lesssim E_C \gg E_J$ is described by the Josephson plasma frequency, given by

$$E_k - E_0 = k\sqrt{8E_M E_C} \, , \tag{22}$$

represented by the red lines. Finally, the limit $E_J, E_M \ll E_C$ is given by Eq. (21).

We note that the ground state remains singly degenerate throughout the reference protocol. This is essential to ensure convergence to a well-defined parity state at $f$. Therefore, all the operations in the protocol have to be adiabatic, leading to a limiting timescale given by Eq. (8). Approximating the numerator in the equation by its maximum value, 1, we get conservative

estimates for the lower timescale limit, given by

$$\Delta t_{i\to 2} \ \gg \ \frac{E_J^{\max} - E_C}{16(E_M^{\max})^2} + \frac{1}{E_C\left[\left(1-\{n_g\}\right)^2 - \{n_g\}^2\right]^2} , \tag{23}$$

$$\Delta t_{2\to f} \ \gg \ \frac{E_M^{\max}}{E_C^2\left[\left(1-\{n_g\}\right)^2 - \{n_g\}^2\right]^2} , \tag{24}$$

for $E_M^{\max} \ll E_C$ and the two steps of in the reference protocol, shown in Fig. 5 (a) and (b). The lower bound for the timescale of the fusion protocol gets reduced when increasing $E_M^{\max}$, as it increases the energy of the lowest excitation energy.

In the opposite regime where $E_M^{\max} \gg E_C$, the limiting timescales are given by

$$\Delta t_{i\to 2} \ \gg \ \frac{\mathrm{Log}\left[E_J^{\max}/\min\left(E_J^{\max}, E_M^{\max}\right)\right]}{8E_C} + \frac{\min\left(E_J^{\max}, E_M^{\max}\right)}{8E_J^{\max}E_C} , \tag{25}$$

$$\Delta t_{2\to f} \ \gg \ \frac{\log\left(E_M^{\max}/E_C\right)}{8E_C} + \frac{E_M^{\max}}{E_C^2\left[\left(1-\{n_g\}\right)^2 - \{n_g\}^2\right]^2} , \tag{26}$$

where we see that the timescale for the reference protocol is minimal for $E_M^{\max} \geq E_J^{\max}$. In this situation, the timescales for the reference and the fusion protocol are similar.

## B  Adiabatic charge evolution

To better illustrate how charge is pumped during the fusion and reference protocols, we analyze the evolution of the average charge in the island for the steps in the two protocols. In Fig. 6, we show the island charge evolution for the fusion (reference) protocol in the panels (a) and (b) ((c) and (d)). For the first step in the fusion protocol, $i \to 1$ in Fig. 3 (a), the island charge is constant when reducing $E_M$. This is due to the large $E_J^{\max}$ chosen, quenching charging effects in the island. In the second step, $1 \to f$, the ground state degeneracy is lifted by reducing $E_J$, as shown in Fig. 3 (c). This also makes the island charge occupation different between the ground and the first excited state, illustrated by the splitting of the solid and dashed lines in Fig. 6 (b). This splitting is due to the different MBS parity states of the ground and excited states. As a result, the island is in an equal superposition between states with $Q$ and $Q+1$ charges at $f$, if the operation is non-adiabatic with respect to the excitation energy to the first excited state, see Sec. 4 for details. We expect that the superposition decays fast into a state with a well-defined island charge, which is the outcome measurement of the protocol.

In the charge pumping protocol, after the island is isolated at $f$, the coupling to the TSs is switched on again, $f \to 2$. This corresponds to following the green arrow backwards in Figs. 3 (a) and Fig. 6 (d). At $f$ the system is in a state with $Q$ or $Q+1$ (or $Q-1$, depending on $n_g$) charges. By opening the tunnel barriers to the outermost TS leads, the excess charge relaxes. In the final step of the pumping protocol, the JJ FET is opened ($2 \to i$), increasing $E_J$, Fig. 6 (c). Due to the large value taken for $E_J$, charging effects are quenched in $i$, leading to a ground state with an undefined number of charges in the island. During this process, some charge flows through the JJ FET.

Alternatively, charging effects can be quenched by the tunneling between the MBSs in the limit $E_M^{max} \gg E_C$, as illustrated by the region to the left of the green dot in Fig. 6 (d). In this case, charge is relaxed by tunneling to the outermost TSs and no charge flows through the JJ FET in step $2 \to f$, panel (d).

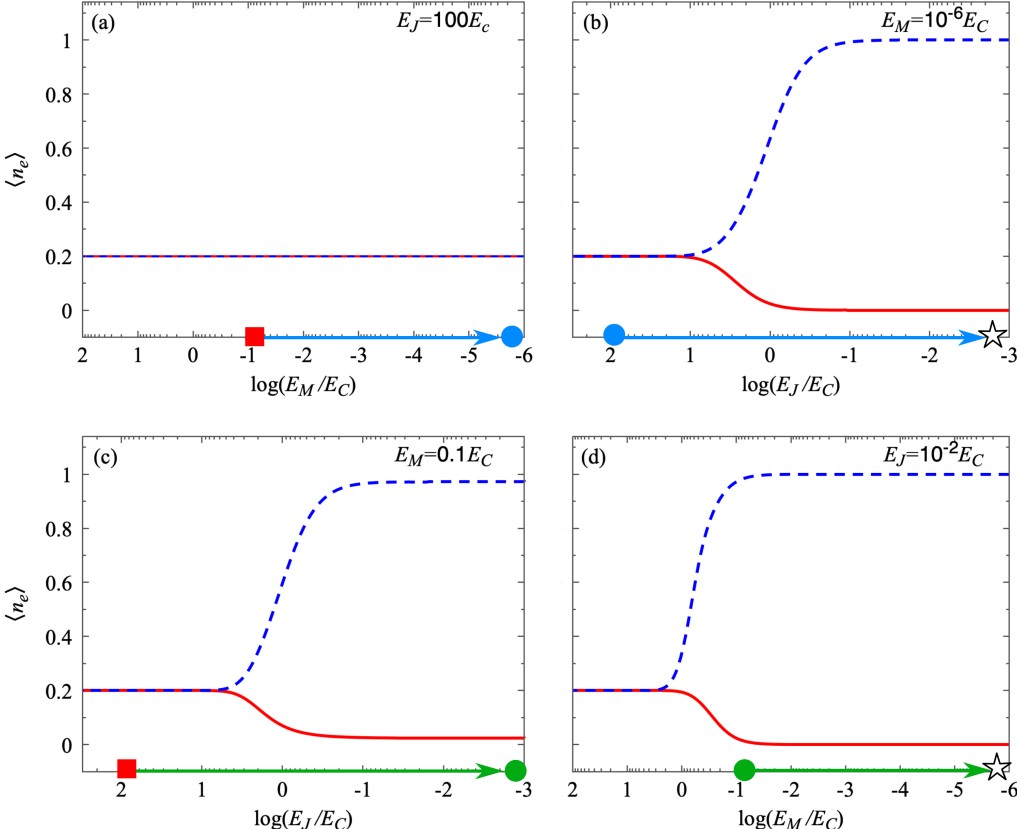

Figure 6: Average charge in island for the system ground (solid red line) and first excited state (dashed blue line). Panels (a) and (b) relate to the fusion protocol, while (c) and (d) relate to the reference protocol. The remaining parameters are the same as in Fig. 3.

The different behavior between the $E_M^{\max} \ll E_C$ and $E_M^{\max} \gg E_C$ limits is better illustrated in Fig. 7 (a) and (b), where we represent the average electron number in the island at $i$, $f$, and 2. For $E_J^{\max} \gg E_C$, the average electron number in 1 (not shown) is the same as the one in $i$. As shown, the ground state charge in $i$ is linear in the excess charge as $\langle n_e \rangle = n_g$. At $f$, the system ideally converges to an equal superposition of the ground and excited state, showing $1e$ steps at integer $n_g$ values, see black and gray curves. Finally, the charge in the ground state at 2 exhibits either steps for the $E_M^{\max} \ll E_C$ limit, Fig. 7 (a), or a linear dependence on $n_g$ for $E_M^{\max} \gg E_C$, Fig. 7 (b).

In these two limits, the charge per cycle pumped through the JJ FET is given by

$$\Delta e = \langle n_e \rangle_f - \langle n_e \rangle_2, \tag{27}$$

where the sub-index denotes the point in parameter space shown in Fig. 3 (a). We have defined $\langle n_e \rangle_f$ as the average charge in the island in the ground and excited state, while $\langle n_e \rangle_2$ corresponds to the ground state one. Here, we have considered that the excess charge is relaxed through the JJ FET in the step $2 \to f$, which is a good approximation in the limits $E_M^{\max} \ll E_C \ll E_J^{\max}$ and $E_M^{\max} \gg E_C$, where charging effects are quenched at 2. The same result is found for the charge pumped to the outermost TSs with an opposite sign.

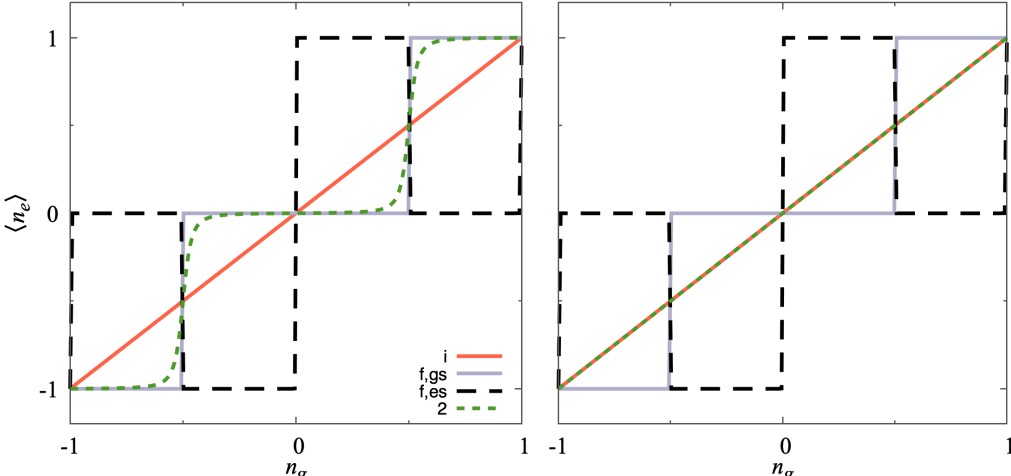

Figure 7: Average number of electrons in the island at $i$, $f$, and 2 steps (see Fig. 3), where $gs$ and $es$ denote the ground and excited states. $\langle n_e \rangle$ in 1 is the same as in $i$ for $E_J \gg E_C$. We show results for $E_M^{\max} = E_C/10$ (b) and $E_M^{\max} = 10E_C$ (a). Here, $E_J^{\max} = 100E_C$, $E_J^{\min} = E_C/100$, and $E_M^{\min} = 10^{-6}E_C$. The vertical axis has been shifted such that the curves are centered at $\langle n_e \rangle = 0$.

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
