# Peer review of "Fusion rules in a Majorana single-charge transistor"

_SciPost Physics, doi:SciPost Phys. 12, 161 (2022)_

## Round 1 · Referee Report · Anonymous (Referee 1) · 2022-1-24

Strengths

1.Proposal for Majorana fusion rules that presents important improvements as compared to previous proposals.

Weaknesses

1. In my opinion, the state-of-the-art concerning Majorana detection is not properly described in the introduction.
2.The low-energy model is somewhat too simplified with important ingredients missing (see report).

Report

This paper presents a novel proposal for Majorana fusion rules that presents important improvements as compared to previous proposals (in particular, the proposed protocol seems simpler than the ones presented in Refs. [33,49] : simpler geometry, less steps, etc.). My overall impression is good and I think the paper deserves to be considered for publication. Before this, however, some issues need to be addressed, in particular:
1) The introduction poorly reflects the state-of-the-art of Majorna detection, since it is somewhat too optimist: sentences like "Despite the increasing experimental evidence consistent with MBSs, the experimental demonstration of the non-abelian and/or non-local properties of MBSs remains an open challenge in the field" seem to suggest that there is enough experimental evidence for MBS detection such that we are ready for the next steps (non-local\non-abelian measurements) which is misleading (in particular for non-experts). The issue of trivial zero energy Andreev states is mentioned in passing and deserves a more thorough discussion (Also, this recent review Prada et al, Nature Reviews Physics 2, 575-594, 2020, should be cited). Other sentences that need clarification include claims such as "detecting the regime where all parts of the structure are in the required TS phase can be done rather straightforward based on a measurement of the supercurrent".
2) The model is way too simplified, with important (minimal) ingredients missing. I am worried, in particular, about the simple island model in Eq. 4, which only includes microscopic Majorana degrees of freedom. In a realistic island, other subgap states may coexist with MBSs spoiling the fusion protocol. This is of importance at not so large magnetic fields (say after the topological transition) where the lowest quasiparticle excitations. Other important simplifications that may affect the conclusions of the paper include the neglect of the outer Majoranas in the grounded SC leads, the assumption that E_M is equal in both junctions (and constant), and the neglect of the Josephson coupling between the grounded SC leads and the floating SC. As an example of the important deviations that a minimal microscopic model of such nw-based islands may induce in the transmon-like levels is discussed in Avila et al, Physical Review B 102, 094518, 2020 and Physical Review Research 2, 033493, 2020 (these two papers deserved to be cited in the context of this paper). Importantly, changes in the spectrum may lead to large deviations in the fusion timescales (Eqs.7-8) and the overall analysis in terms of energy scales in section 4. Can the authors comment on this?
I am not asking the authors to do extra calculations but a detailed discussion about the limitations of their simplifications various places of the paper is in order.
-Finally, and in relation to my first point about trivial zero modes versus MBSs, do the authors expect that some of these nontrivial fusion rules would survive in the presence of non-topological zero modes? ( Note that some previous papers have already discussed how non-topological zero modes also exhibit non-Abelian properties: Vuik et al, SciPost Phys. 7, 61, 2019; San Jose et al Scientific reports 6, 1-13, 2016). I think this is an important discussion in view of the state of affairs concerning Majorana detection in nanowires.

  • validity: good
  • significance: good
  • originality: good
  • clarity: high
  • formatting: excellent
  • grammar: excellent

Author:  Rubén Seoane Souto  on 2022-03-02  [id 2259]

(in reply to Report 1 on 2022-01-24)
Category:
answer to question

We thank the Referee for attentively reading the manuscript and for his/her positive comments.

1) We have re-written the introduction, avoiding too optimistic sentences, as suggested by the Referee. We have furthermore extended the discussion about trivial zero-energy modes, including the reference mentioned by the Referee.

We agree with the referee that the sentence he mentioned can be confusing. In fact, we have a draft manuscript where we investigate transport through this system and had expected that to appear on the arXiv shortly after the present paper. However, for various reasons the transport paper has been delayed and we therefore decided to remove this statement, because referring to an unpublished work is not very helpful. In short, our still unpublished results show that the supercurrent through the system is significantly enhanced after the topological transition. We also show how to infer the energy of the bound states in the different parts of the system using supercurrent measurements.

2) The presence of other subgap states can modify the limiting timescales for the fusion and the reference protocols. Our proposal is based on the adiabatic condition, requiring that the system remains in its ground state (except close to $1$, where it is almost degenerate). Therefore, quasiparticle excitations involving trivial states are detrimental for our proposal. We have included a comment about this possible issue on page 11.

In our work, we have considered that the lead wires are sufficiently long such that the outermost MBSs do not couple to the 4 inner ones. In this limit, the outermost MBSs don't contribute to the fusion outcome. If there is a finite coupling to the outermost MBSs (or to any other near-zero states in the leads), the protocol should be carried out faster than the rate associated with that coupling. We have added a comment about it at the end of section 4.

We note there is no requirement for $E_M$ to be symmetric. The fusion (reference) protocols are based on the degenerate (non-degenerate) ground state. This property does not depend on how $E_M$ is switched off.

The Josephson coupling between the grounded and the island superconductors tends to increase the energy splitting between the ground and the excited states (except when the island decouples from the leads). It, therefore, helps at reducing the lower timescale limit, increasing the time range where fusion rules can be demonstrated. We have neglected this term in order to get analytic conservative estimates for the fusion timescale limits. We have included a reference to the mentioned articles and a comment about the effect of $E_J$ between the leads and the island.

We agree with the Referee that changes in the spectrum when switching the different gates can affect the protocol outcome if trivial states (either subgap or at the gap) approach zero energy. For this reason, it is important to tune the device in such a way that no trivial states come close to zero energy. We have included a comment in the manuscript.

3) The fusion signal is based on the ground state degeneracy at point $1$ in Fig. 3 (a). A trivial state can be decomposed as two MBSs at the same end of the wire with, possibly, a small overlap integral. However, the two MBS components can split due to the tunneling to superconductor/s. In general, it will split the ground state degeneracy. Therefore, if operations are adiabatic, our proposal will be able to distinguish between trivial states and MBSs. However, if the inverse operation time is much larger than the ground state splitting at $1$, trivial states will provide the same outcome as the topological states after our fusion experiment. This is particularly true if one of the MBS components do not couple to the other wires. As mentioned by the Referee and in the references he/she mentioned, these trivial states can reproduce a full range of topological signatures, including non-abelian properties. Similar limitations apply to other fusion and braiding proposals.

We have added a comment and a reference to the mentioned articles in the introduction and the conclusions.

---

## Round 1 · Referee Report · Elsa Prada (Referee 2) · 2022-2-1

Report

In this work the authors propose and theoretically analyze a way to demonstrate Majorana bound state (MBS) fusion rules in a Majorana single-charge transistor. Their proposal presents several advantages with respect to previous ones.

I have to say that I was ignorant of the mechanism and proposals that had dealt with this particular subject (fusion rules in Majorana wires) over the last years, and this manuscript has really illuminated me. It is very well written, it provides an excellent motivation to the subject, a clear, simple and concise explanation of the physics involved, and a context and critical comparison of their proposal with others in the literature.

To analyze this problem, the authors use a low energy model with many simplifications, in order to gain insight and to be able to obtain analytical expressions for the operation timescales. My only criticism to this type of studies is that we have learnt over the years how much more complicated Majorana nanowires are with respect to these type of simplified models, and how many things can "go wrong" in realistic systems/microscopic models to try to create and manipulate MBSs. Nevertheless, I consider these studies very necessary as a proof-of-principle-type of analysis, and as an important guidance for experimentalists for the future, when the Majorana nanowires improve their quality and we achieve good control over them. Moreover, due precisely to the simplicity of the model, I think the authors are able to calculate and discuss in an insightful way the phenomenology of the protocol, the relevant energy scales, the different upper and lower bounds for the operation timescales and the sources of decoherence/noise.

Thus, I find the manuscript of importance to the field, well written, the calculations look sound, and the results well explained and discussed. For these reasons, I recommend this work for publication in SciPost. I just have some small comments and questions:

1) I have to say that I find a couple of references missing in the introduction. When the authors enumerate the experimental measurements compatible with the existence of Majoranas, they mention "the hybridization with quantum dot orbitals", and they very rightly cite Ref. [16]. I think the authors could also introduce there a reference to Phys. Rev. B 98, 085125 (2018):
https://journals.aps.org/prb/abstract/10.1103/PhysRevB.98.085125
On the other hand, when the authors say "Without such a demonstration, it seems impossible to distinguish with certainty true topological MBSs from different types on non-topological states that might also explain many of the experimental observations so far". I think that in the list of references they provide here, the authors could add a recent review where exactly this subject has been exhaustively discussed: Nature Reviews Physics 2, 575–594 (2020)
https://www.nature.com/articles/s42254-020-0228-y

2) Can the authors comment what would be the effect of including the outer Majoranas of the lead topological superconductors (i.e., having 6 Majoranas instead of 4 in the low energy model)? Would this simply modify the bounds for the operation timescales or would they destroy the fusion protocol altogether?

3) In the same spirit, do the author have an opinion on what would happen if instead of true Majoranas, one would have quasi-Majoranas, say, for instance, in the superconducting leads at the left and right of the TS island? Could these "mimic" the results for the topological fusion rules? I'm not asking for a calculation, but just an informed guess.

4) Finally, since an important point of this paper is that the authors propose a simplified device/scheme for the detection of fusion rules with respect to previous proposals (as listed at the beginning of page 4), how do the final time scales found (the ones given in the conclusions) compare to those previous proposals? Are they better, more flexible? If yes, perhaps this could be briefly mentioned or highlighted in the conclusions.

  • validity: -
  • significance: -
  • originality: -
  • clarity: -
  • formatting: -
  • grammar: -

Author:  Rubén Seoane Souto  on 2022-03-02  [id 2260]

(in reply to Report 2 by Elsa Prada on 2022-02-01)
Category:
answer to question

We thank the Referee for attentively reading our manuscript and the positive recommendation.

1) We thank the Referee for bringing to our attention the missing relevant references that are now included in the revised manuscript.

2) In our work, we have considered that the lead wires are sufficiently long such that the outermost MBSs do not couple to the 4 inner ones. In this limit, the outermost MBSs don't contribute to the fusion outcome. If there is a finite coupling to the outermost MBSs (or to any other near-zero states in the leads), the protocol should be carried out faster than the rate associated with that coupling. We have added a comment about it at the end of section 4.

3) The fusion signal is based on the ground state degeneracy at point $1$ in Fig. 3 (a). A trivial state can be decomposed as two MBSs at the same end of the wire with, possibly, a small overlap integral. However, the two MBS components can split due to the tunneling to superconductor/s. In general, it will split the ground state degeneracy. Therefore, if operations are adiabatic, our proposal will be able to distinguish between trivial states and MBSs. However, if the inverse operation time is much larger than the ground state splitting at $1$, trivial states will provide the same outcome as the topological states after our fusion experiment. This is particularly true if one of the MBS components do not couple to the other wires. These trivial states can reproduce a full range of topological signatures, including non-abelian properties. Similar limitations apply to other fusion and braiding proposals.

We have added a comment and a reference to the mentioned articles in the introduction and the conclusions.

4) In the limit of clean system and large gap, the protocol can be carried out in a timescale $\sim 10^{-10}$ s. In this situation, experiments will be likely limited by the time response of electronic components.

If superconductors feature trivial low-energy states, the fusion timescale will increase, limited by transitions to these other states. Our simpler protocol will likely lead to somewhat shorter timescales thanks to the reduced number of operations. However, the real constraints are probably set by various experimental limitations and we prefer not to speculate about this in the paper.

---

## Round 2 · Referee Report · Elsa Prada (Referee 2) · 2022-3-19

Report

The authors have answered satisfactorily the comments/questions raised and they have improved accordingly the manuscript. I think this work is ready for publication in SciPost.

---

## Round 2 · Referee Report · Anonymous (Referee 1) · 2022-3-21

Report

The authors have followed my recommendations and improved the manuscript accordingly. In my opinion, the paper is ready for publication.

---

## Round 2 · List of Changes

We have extended the discussion about trivial states in the introduction and the conclusions.
We have included more details about how to measure zero-energy states in our geometry through transport.
We have removed point (5) in page 4.
We have included a comment about the effect of trivial states in the protocol at the end of section 4.
We have added Refs. [7,18,22-25,71,72].

---

## Editorial Decision

published